# Attitudes towards Statistics among Business Students: Do Gender, Mathematical Skills and Personal Traits Matter?

**Leiv Opstad** 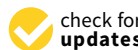

NTNU Business School, Norwegian University of Science and Technology, 7491 Trondheim, Norway;
leiv.opstad@ntnu.no

**Abstract:** The purpose of this article was to investigate different variables, by combining mathematical skills and personal traits using The Big Five Model, to see which have the most influence on business students' attitudes towards statistics. The Big Five personality traits make up a model for capturing various personal characteristics. Specifically, we aimed to understand why there is a gender difference in attitudes towards statistics. Statistical skills are a key factor for success in business studies. The chosen methods were pairwise comparisons (*t*-test) and a linear regression model, using a sample from a Norwegian business school. The finding was that there is a substantial gender gap towards attitudes towards statistics but taking mathematical skills and personal characteristics into consideration then results in this gap becoming much smaller. Furthermore, mathematical skills and personal traits were shown to have an impact on students' attitudes towards statistics.

**Keywords:** attitudes towards statistics; Big Five; mathematical skills; quantitative analysis; gender differences; business studies

---

## 1. Introduction

Attitudes towards statistics influence students' learning in this subject [1]. Business statistics is an important tool in quantitative business courses and especially in courses for a finance major [2]. Many students' quantitative skills do not meet the expectations of the instructors in these fields. Good analysis requires statistical skills. In a subject like business law, there is also an advantage to having statistical abilities [3]. By allowing students to use statistical methods to analyze real data, one can improve students' statistical skills. According to Gal and Garfield [4], students' attitudes towards statistics are particularly important since statistical thinking and application are useful tools in many contexts. Hence, attitudes towards statistics matter for undergraduates in business fields.

The topic of gender gap in higher education continues to attract interest among researchers worldwide. Students' decisions about their educational pathways and careers depend on academic abilities and preferences, and there is a significant gender difference in these preferences [5]. For example, women choose, to a large extent, to study humanities, languages, and social sciences, while sciences appeal more to men. Worthington and Higgs [6] reported that women undergraduates are less likely to continue to study finance after completing introductory courses, preferring instead accounting, marketing, and management. Many women find finance irrelevant or uninteresting. One reason might be that women students are less interested due to their attitudes towards statistics and mathematics [6,7]. Males tend to outperform females in quantitative subjects. The findings in this research contribute to explaining this result.

Economics and business studies attract both men and women. In this field, the gender distribution in Norway is approximately fifty-fifty. Prior research has shown that there is a substantial gender gap

among business and economics students in performance, attitudes, and choice of major [8–10]. One reason might be the mathematical abilities and attitudes towards mathematics [11] This phenomenon is called the confidence gap [12]. Despite gender equalization in many countries, there is still a significant gender difference in mathematical skills [13]. This is due to women tending to choose practical mathematics in high school, while their counterparts prefer theoretical mathematics to a far greater extent. This fact has implications in their success in the different subjects at business school [14].

There is a link between mathematics and statistics [15]. Students who are insecure and have no enthusiasm for mathematics tend to have the same feelings and attitudes towards statistics, according to Ramos Salazar [16]. A problem is that many students do not realize the importance of statistics in business courses [17]. Therefore, the gender gap in mathematics appears also in statistics [16,18]. Women students are less motivated in business statistics than are their men peers. Women and men have different attitudes towards statistics. Statistical skills are important tools in business courses, not only in quantitative courses, but also in subjects such as marketing. In finance courses, statistical abilities are critical for conducting good analyses. A solid background in business statistics leads to better performance and improved careers [19]. Despite its usefulness, many undergraduates struggle with the subject and it brings negative anxiety and antipathy [20,21]. Therefore, it is important to investigate business students' attitudes towards statistics, and especially to understand if there is a gender gap and if so, why. It may help explaining why more women tend to fall behind in some business courses. The purpose of this article is to determine more on this issue.

We conducted this analysis by taking into consideration the mathematical abilities and personality characteristics using the Big Five Model. The question is, after taking these above facets into consideration, is there still a gender difference in attitudes towards statistics, or does it disappear when adjusting for differences in skills and personal traits.

## 2. Theory and Research Model

It is important for business graduates to have analytical skills [22]. Lack of interest, low confidence, and negative attitudes towards statistics can cause poor performance in business courses.

### 2.1. Attitudes towards Statistics

Researchers have used different questions in surveys to measure attitudes towards statistics [1]. Many researchers prefer to use the Survey of Attitude Toward Statistics (SATS-36) developed by Schau et al. [23]. SATS-36 consists of 36 items and has six components: Affect (6 items), Cognitive Competence (6 items), Value (9 items), Difficulty (7 items), Interest (4 items), and Effort (4 items). Affect measures the feelings (positive or negative) about statistics. Cognitive Competence is about intellectual knowledge and skills in using statistics, which corresponds to Self-Concept in attitudes towards mathematics. Value includes relevance—the usefulness and value of statistics. Difficulty measures attitudes about the level of difficulty of statistics as a subject. Interest measures the degree of interest in statistics, and finally Effort gives an indicator of how much time and energy the respondent spends learning statistics. We quantified the responses to the different statements using a Likert scale. All components except for Difficulty reflected positive attitudes in the scores. Following the definition of Schau et al. [23], a high score in the Difficulty category means that the respondent finds statistics easy to learn, and a low score implies that the participant considers statistics to be difficult [22]. One item under this category is "Statistics formulas are easy to understand." A high score here means that the respondent agrees with this statement.

The evaluation of attitudes toward statistics depends on a proper instrument. SATS-36 has powerful reliability and validity [24,25]. However, some authors have recommended removing some items from SATS-36 [26]. SATS-36 is used in this paper.

### 2.2. Gender and Attitudes towards Statistics

Hommik and Luik [27] reported a gender difference in attitudes toward statistics. Men students had marginally higher values in the dimensions of Competence, Value, and Interest, while women students had higher scores in Effort. Rejón-Guardia et al [28] confirmed this tendency with almost the same conclusion. Women had significantly lower mean scores in Value, Competence, and Affect, but higher scores in Effort. The only difference between these two papers is that Affect was substituted for Interest. While Hommik and Luik [27] reported higher scores for men in Interest, Rejon-Guardia et al [28] did not register any gender difference for this factor, but for the dimension Affect, which scored in favor of men. Hannigan et al. [29] also found a gender gap with higher scores for men in Competence and Affect. The other factors had almost even scores for men and women. According to Fullerton and Umphrey [30], women are significantly more anxious than men. Others confirmed that men express more positive attitudes towards statistics [12,31,32]. Women have been shown to be less confident using statistics and to find statistics difficult, and these factors have led to women having negative feelings about the subject. However, the findings have been mixed. Sarikaya et al. [33] and Wisenbaker and Scott [34] did not find any gender difference, and Mahmud and Zainol [35] concluded that women had more positive attitudes toward statistics than men.

### 2.3. Mathematical Skills and Attitudes towards Statistics

Mathematical-orientated students have positive attitudes towards statistics [36]. Note, Chiesi, and Primii [12] did not find that women's negative attitudes towards statistics were related to differences in mathematical knowledge. Hence, it is not obvious how mathematical skills will affect the gender difference in attitudes towards statistics.

### 2.4. The Big Five Personality Traits

The Model Big Five for measuring personal characteristics [37] is widely used all over the world, and many published articles have used this model without conducting any reliability or validity test [38]. It consists of five factors: Agreeableness, Conscientiousness, Neuroticism, Extraversion, and Openness (see Table 1).

**Table 1.** The Big Five.

| Trait | Definition |
| --- | --- |
| Openness to experience | People who have fantasies are open to new experiences and ideas. Example item: I have a lively imagination |
| Conscientiousness | Conscientious people have these personal characteristics: organized, responsible, self-disciplined, effective, and/or goal oriented. Example item: I am always prepared. |
| Extraversion | Extraverts are social and oriented toward the outer world Example item: I am interested in people. |
| Agreeableness | Agreeable people act cooperatively, show trust, and have unselfish manners. Example item: I like to cooperate with others. |
| Neuroticism (inverse of emotional stability) | Neurotic people tend to be emotionally unstable, often have anxiety and/or are depressed. Example item: I get nervous easily. |

### 2.5. Big Five Model and Gender Difference

Many previous researchers [39] have investigated gender differences in personal traits. In line with other research [40,41], Weisberg et al. [42] reported higher scores for women in Extraversion, Agreeableness, and Neuroticism and lower scores than men for Openness and Conscientiousness, with significant differences. This means that the average characteristics of personality for men and women are systematically different. According to Costa et al. [40] the gender gaps in Neuroticism and

Agreeableness are stable due to biological gender differences. Chapman et al. [43] pointed out that men scored higher in some items/aspect of extraversion while women achieved higher scores in others. All together they found a moderately higher value for men than women in the dimension of Extraversion. Using the Big Five Inventory with samples from 55 nations (N = 17,637) Schmitt et al. [39] reported that women had higher scores in Neuroticism, Extraversion, Conscientiousness, and Agreeableness across most countries and cultures.

The model in this study is to see the level of differences in attitudes towards statistics based on gender, mathematical skills, and personal characteristics.

### 2.6. Hypotheses

Based on the literature review, we postulate the following hypotheses:

**Hypothesis 1 (H1):** *There is a gender difference in personal characteristics among business students.*

**Hypothesis 2 (H2):** *There is a gender difference in attitudes towards statistics among business students.*

**Hypothesis 3 (H3):** *Mathematical skills and personal characteristics have influence on students' attitudes towards statistics.*

**Hypothesis 4 (H4):** *The gender difference in attitudes towards statistics gets smaller when taking differences in mathematical skills and personal traits into account.*

Previous research has documented that there is a gender gap in attitudes towards statistics and in personal traits. We wondered if there are gender differences using data from a Norwegian business school.

We are not aware of any previous study that has analyzed the gender difference in attitudes towards statistics by linking this topic to mathematical skills and personal characteristics. Since personal traits and mathematical abilities vary by gender, the assumption is that the gender distinction in attitudes towards statistics decreases when these factors are taken into account (H4). Previous studies have shown that mathematical skills and personal characteristics influence attitudes towards statistics [29,44]. Therefore, we postulate hypothesis H3.

## 3. Sample and Research Methodology

### 3.1. Sample

The sample consisted of approximately 140 students per year for the year 2019. The questionnaire was spread among undergraduates attending the second-year compulsory macroeconomics course on a particular day in the Fall semester. The data might be marginally biased, since around 30% of the students were generally absent on the day of the survey. Nevertheless, the survey gives a picture of those who chose to attend the lectures on those days.

The instruments for measuring the attitudes towards statistics (SATS-36) and personal traits are based on methods applied in international studies. Some researchers have used a modification of SATS-36 [24,45]. In this study, we used the original version of SATS-36 (Attitudes towards statistics) developed by Schau et al. [23]. The 20-item version of the Big-Five Inventory (BFI-20) is identical to the one developed by Engvik and Clausen [46]. The Appendix A provides a more detailed descriptive statistical overview (Tables A1 and A2).

### 3.2. The Model

The Model's Dependent Variable: Attitudes towards Statistics Using a linear regression model enabled us to simultaneously find how attitudes towards statistics depend on gender, mathematical skills, and personal traits (See Figure 1). We used the following model:

$$Yi = a_0 + a_1X1 + a_2 X2 + a_3X3 + a_4X4 + a_5X5 + a_6X6 + a_7X7 + \varepsilon$$

where

$Yi$: Factor $i$ attitudes towards statistics, $i$ = Affect, Value, Difficulty, Interest, Cognitive Competence or Effort (Likert scale 1 to 7)

$\alpha_0$: Constant

$X_1$: Gender (0:F, 1:M)

$X_2$: Dummy variable for N-mathematics (0: Non N-maths, 1: N-maths)

$X_3$: Dummy variable for S-mathematics (0: Non S-maths, 1: S-maths)

$X_4$: Agreeableness (Likert scale 1 to 5)

$X_5$: Conscientiousness (Likert scale 1 to 5)

$X_6$: Extraversion (Likert scale 1 to 5)

$X_7$: Neuroticism (Likert scale 1 to 5)

$\varepsilon$: stochastic error

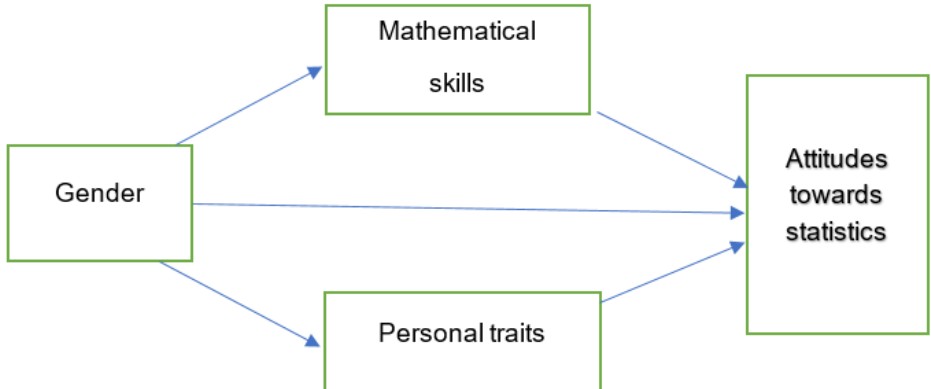

**Figure 1.** Research model: Variables influencing attitudes towards statistics.

A method for distinguishing between direct and indirect gender effects is to use mediation analyses [47,48]. An alternative approach is to have different sets of variables in the regression model [49–51]. In step 1 only gender enters as background variable in this stepwise regression model, step 2 includes gender and personal traits as independent variables, and finally step 3 with the complete model. Step 1 shows the correlation between gender attitudes towards statistics while step 3 reveals the direct effect. The difference between step 1 and step 3 implies the indirect effect (of personal traits and mathematical skills).

Unfortunately, there was no access to experimental data. Although there is correlation between the independent variables and the dependent variable, one should therefore be careful to draw conclusions about causal relationships. To avoid multicollinarity, the dummy variable for P- mathematics was removed from the regression model and this group belongs to the reference category, which also includes some other students.

### 3.3. The Data

The value of the dimensions for the Big Five (using a 5-point Likert scale) varied between 2.7 and 3.9, with lowest score for Neuroticism and the highest for Agreeableness (Table 2a).

**Table 2.** Descriptive statistics.

| a. Data from the Big Five Model, Personal Traits (N = 130). | | | | |
|---|---|---|---|---|
| Factor | Mean | Min | Max | Std. Deviation |
| Agreeableness | 3.91 | 225 | 5.0 | 0.57 |
| Conscientiousness | 3.67 | 1.50 | 5.0 | 0.70 |
| Extraversion | 3.64 | 1.75 | 5.0 | 0.78 |
| Openness | 3.33 | 1.50 | 5.0 | 0.73 |
| Neuroticism | 2.70 | 1.0 | 4.50 | 0.79 |
| b. Data from Attitudes Toward Statistics (N = 139, 1 year, 2019). | | | | |
| Factor | Mean | Min | Max | SD * |
| Affect | 4.60 | 1.0 | 7.0 | 1.25 |
| Value | 4.45 | 1.33 | 6.67 | 1.00 |
| Difficulty (find statistics easy) | 3.52 | 1.14 | 5.29 | 0.75 |
| Interest | 4.51 | 1.0 | 7.0 | 1.29 |
| Cognitive Competence | 5.19 | 1.83 | 7.0 | 1.08 |
| Effort | 5.67 | 1.50 | 7.0 | 1.09 |
| c. Mathematics Skills. The business students' choice of mathematical pathway at the Secondary-Upper School (N = 129). | | | | |
| Mathematics | Women | Men | All | |
| Practical mathematics (P-maths) | 13 (59.1%) | 9 (40.9%) | 22(100%) | |
| Mathematics for business and social science (S-maths) | 35 (53.8%) | 30 (46.5%) | 65 (100%) | |
| Mathematics for natural science (N-maths) | 14 (36.8%) | 24 (63.2%) | 38 (100%) | |
| Others | 4 (100%) | 0 (0%) | 4 (100%) | |
| All | 66(51.4%) | 63 (48.6%) | 129 (100%) | |

* SD = standard deviation.

There was also a considerable variation in the attitudes towards statistics (Table 2b). On a 7-point Likert scale, Difficulty was at the bottom with a value of 3.52 and Effort was on top with a value of 5.67. There were also some differences in the values of the standard deviation.

Table 2c shows that the majority of business undergraduates are women (51.2%). Notice also that most of the women students preferred practical mathematics (P-maths) or mathematics for business and social science (S-maths). Among the students with backgrounds in mathematics for natural science (N-maths), there was an underrepresentation of women. More men than women choose N-mathematics at the upper secondary school level. The focus in practical mathematics (P-maths) is to teach the students about useful functions for making mathematical models of practical relationships. The subjects in S-mathematics include algebra, exponential functions, and regression analysis. The students deal with symbols and use of formulae. N-mathematics contain calculations, and analysis of figures in a plane. A main field is geometry including the use of vectors.

## 4. Finding

### 4.1. Descriptive Gender Differences and Pairwise Comparison Using T-Test

This study shows a gender difference based on the Big Five personal traits. Women's scores were higher than their men peers in the traits of Agreeableness, Conscientiousness, and Neuroticism. Only for the dimension of Openness did men get higher scores than women, but the correlation was not significant.

The independent sample test of mean verified that there are gender differences in personal characteristics. The validity was under the 1% significance level for Neuroticism, and under 10% for Agreeableness and Conscientiousness. The results confirmed hypothesis H1.

There was also a gender gap in attitudes towards statistics in which the men tended to have higher scores (Table 3b). For the factors Value, Difficulty (find statistics easy), and Interest, the differences were strongly significant on the 1% level. However, the effect was weaker for Affect and Cognitive

Competence, but still significant. On the other hand, for Effort there was a substantial gender difference in favor of women (higher score).

**Table 3.** Gender differences, independent *t*-test.

| **a. Independent Sample *t*-Test, Big Five, Personal Traits (Assume Equal Variance).** | | | | | |
|---|---|---|---|---|---|
| Factor | Women | Men | Difference | *t*-Value | Significance Level (2-tailed) |
| Agreeableness | 4.00 | 3.81 | 0.19 (0.10) | 1.91 | 0.058 |
| Conscientiousness | 3.79 | 3.59 | 0.20 (0.10) | 1.71 | 0.09 |
| Extraversion | 3.64 | 3.62 | 0.02 (0.14) | 0.064 | 0.95 |
| Openness | 3.30 | 3.36 | −0.06 (0.13) | −0.44 | 0.66 |
| Neuroticism | 3.05 | 2.33 | 0.72 (0.12) | 5.81 | 0.000 |
| **b. Independent Sample *t*-Test, Gender and Attitudes Towards Statistics (Assume Equal Variance).** | | | | | |
| Factor | Women | Men | Difference | *t*-Value | Significance Level (2-tailed) |
| Affect | 4.43 | 4.81 | −0.39 (0.22) | −1.76 | 0.081 |
| Value | 4.22 | 4.70 | −0.49 (0.17) | −2.83 | 0.005 |
| Difficulty (find statistics easy) | 3.32 | 3.75 | −0.43 (0.13) | −3.32 | 0.001 |
| Interest | 4.19 | 4.87 | −0.67 (0.22) | −3.10 | 0.002 |
| Cognitive Competence | 5.05 | 5.38 | −0.33 (0.19) | −1.73 | 0.086 |
| Effort | 5.97 | 5.38 | 0.59 (0.19) | 3.16 | 0.0028 |

Note. Standard deviation in parenthesis.

The independent sample *t*-test of mean showed that there are gender differences in attitudes towards statistics. The data confirmed hypothesis H2.

### 4.2. Result from the Regression Model

The regression analysis indicated that mathematical background is related to attitudes towards statistics (Table 4a,b), but not for all factors. For N-mathematics, there were positive significant links for Affect ($\beta = 0.67$, $p < 0.05$), Value ($\beta = 0.34$, $p < 0.10$), and Difficulty ($\beta = 0.32$, $p < 0.05$). Furthermore, there was a negative connection between S-mathematics and Effort ($\beta = -0.42$, $p < 0.10$). There was also a link between personal characteristics and attitudes towards statistics. Neuroticism had a negative correlation to the dimensions of Affect and Cognitive Competence, while Openness was negatively related to Cognitive Competence and positively related to Interest, both effects being significant. The findings confirmed hypothesis H3, that mathematical skills and personal characteristics are linked to students' attitudes towards statistics.

Note, the scores of the regression coefficient (β) decrease from step 1 to step 3 in the regression models (Table 5). Taking into consideration the indirect effect of mathematical skills and personal traits, the correlations between gender and attitudes towards statistics appear to be smaller. This effect is substantial for Cognitive Competence Affect and Effort. There is no longer any significant correlations for those three factors. The change is also extensive for Value (from 0.5% to 8.9%) For the other two factors, Difficulty and Interest), the inclusions of mathematical skills and Big Five have a minor effect on the significance level. The pairwise comparison illustrates the same picture (Table 3b), where the *t*-test showed also a strong significant gender difference. The finding (Table 5) confirms hypothesis H4, that the gender difference in attitudes towards statistics became smaller when taking into account the differences in mathematical skills and personal traits.

**Table 4.** Results from stepwise multiple regression analyses.

| | a. Results of the Regression Model: Attitudes towards Statistics. (Unstandardized Coefficients, $\beta$). | | | | | | | |
|---|---|---|---|---|---|---|---|---|
| | **Affect** | | **Value** | | **Difficulty (Easy)** | | **Cognitive Competence** | |
| | **Coefficient $\beta$** | **Significance Level** | **Coefficient $\beta$** | **Significance Level** | **Coefficient $\beta$** | **Significance Level** | **Coefficient $\beta$** | **Significance Level** |
| **Step 1** | | | | | | | | |
| Constant | 4.41 | | 4.22 | | 3.32 | | 5.04 | |
| Gender | 0.41 (0.22) | 0.066 | 0.49 | 0.005 | 0.43 (0.13) | 0.001 | 0.34 (0.19) | 0.074 |
| | N = 127 | | N = 127 | | N = 127 | | N = 127 | |
| | Adj. $R^2$ = 0.019 | | Adj. $R^2$ = 0.052 | | Adj. $R^2$ = 0.073 | | Adj. $R^2$ = 0.017 | |
| **Step 2** | | | | | | | | |
| Constant | 5.01 | | 4.05 | | 4.00 | | 6.45 | |
| Gender | 0.15 (0.25) | 0.58 | 0.40 (0.20) | 0.045 | 0.36 (0.15) | 0.019 | 0.059 (0.21) | 0.29 |
| Agreeableness | −0.08 (0.20) | 0.78 | −0.17 (0.16) | 0.30 | −0.16 (0.12) | 0.20 | −0.28 (0.17) | 0.10 |
| Conscientiousness | 0.40 (0.17) | 0.022 | 0.32 (0.14) | 0.021 | 0.11 (0.10) | 0.30 | 0.40 (0.14) | 0.006 |
| Extraversion | 0.042 (0,15) | 0.78 | −0.08 (0.12) | 0.50 | 0.10 (0.09) | 0.29 | 0.071 (0.13) | 0.58 |
| Openness | −0.19 (0.17) | 0.26 | 0.12 (0.14) | 0.38 | −0.15 (0.10) | 0.13 | −0.22 (0.14) | 0.12 |
| Neuroticism | −0.45 (0.17) | 0.009 | −0.16 (0.14) | 0.25 | −0.10 (0.10) | | −0.44 (0.14) | 0.002 |
| | N = 127 | | N = 127 | | N = 127 | | N = 127 | |
| | Adj. $R^2$ = 0.094 | | Adj. $R^2$ = 0.073 | | Adj. $R^2$ = 0.081 | | Adj. $R^2$ = 0.137 | |
| **Step 3** | | | | | | | | |
| Constant | 4.76 | | 3.80 | | 3.82 | | 6.32 | |
| Gender | 0.056 (0.24) | 0.82 | 0.34 (0.20) | 0.089 | 0.32 (0.15) | 0.035 | 0.002 (0.20) | 0.99 |
| N-maths | 0.67 (0.31) | 0.03 | 0.51 (0.25) | 0.044 | 0.32 (0.19) | 0.097 | 0.36 (0.26) | 0.17 |
| S -maths | −0.12 (027) | 0.66 | −0.04 (0.22) | 0.87 | 0.07 (0.17) | 0.68 | −0.20 (0.23) | 0.39 |
| Agreeableness | −0.07 (0.19) | 0.72 | −0.16 (0.16) | 0.30 | −0.16 (012) | 0.19 | −0.26 (0.16) | 0.11 |
| Conscientiousness | 0.37 (0.17) | 0.03 | 0.30 (0.16) | 0.028 | 0.10 (0.10) | 0.35 | 0.38 (0.14) | 0.008 |
| Extraversion | 0.11 (0.15) | 0.45 | −0.04 (0.12) | 0.78 | 0.12 (0.09) | 0.19 | 0.12 (0.16) | 0.77 |
| Openness | −0.21 (0.16) | 0.19 | 0.10 (0.13) | 0.44 | −0.16 (0.10) | 0.12 | −0.26 (0.14) | 0.087 |
| Neuroticism | −0.41 (0.16) | 0.01 | −0.13 (0.13) | 0.35 | −0.08 (0.10) | 0.42 | −0.42 (0.14) | 0.003 |
| | N = 127 | | N = 127 | | N = 127 | | N = 127 | |
| | Adj. $R^2$ = 0.16 | | Adj. $R^2$ = 0.12 | | Adj. $R^2$ = 0.09 | | Adj. $R^2$ = 0.18 | |

**Table 4.** *Cont.*

| | Effort | | Interest | |
|---|---|---|---|---|
| **b. Results of the Regression Model: Attitudes towards Statistics. (Unstandardized Coefficients β).** | | | | |
| | | **Dependent Variable** | | |
| | **Effort** | | **Interest** | |
| | **Coefficient β** | **Significance Level** | **Coefficient β** | **Significance Level** |
| **Step 1** | | | | |
| Constant | 5.97 | | 4.17 | |
| Gender | −0.59 (0.19) | 0.002 | 0.70 | 0.002 |
| | N = 127 Adj. $R^2$ = 0.066 | | N = 127 Adj. $R^2$ = 0.117 | |
| **Step 2** | | | | |
| Constant | 1.66 | | 2.71 | |
| Gender | −0.24 (0.20) | 0.23 | 0.61 (0.25) | 0.015 |
| Agreeableness | 0.25 (0.16) | 0.13 | −0.11 (020) | 0.59 |
| Conscientiousness | 0.55 (0.14) | 0.00 | 0.44 (0.17) | 0.011 |
| Extraversion | 0.11 (0.12) | 0.39 | −0.18 (0.15) | 0.23 |
| Openness | 0.006 (0.14) | 0.96 | 0.42 (0.17) | 0.014 |
| Neuroticism | 0.27 (0.14) | 0.052 | −0.15 (0.17) | 0.36 |
| | N = 127 Adj. $R^2$ = 0.22 | | N = 127 Adj. $R^2$ = 0.094 | |
| **Step 3** | | | | |
| Constant | 1.93 | | 2.53 | |
| Gender | −0.21 (0.20) | 0.29 | 0.56 (0.25) | 0.025 |
| N-maths | −0.37 (0.26) | 0.15 | 0.35 (0.32) | 0.27 |
| S-maths | −0.42 (0.23) | 0.07 | 0.02 (0.29) | 0.96 |
| Agreeableness | 0.27 (0.16) | 0.09 | −0.11 (0.20) | 0.59 |
| Conscientiousness | 0.56 (0.14) | 0.00 | 0.43 (0.17) | 0.01 |
| Extraversion | 0.11 (0.12) | 0.40 | −0.15 (0.16) | 0.33 |
| Openness | 0.004 (0.14) | 0.98 | 0.41 (0.17) | 0.02 |
| Neuroticism | 0.24 (0.14) | 0.08 | −0.13 (0.17) | 0.43 |
| | N = 127 Adj. $R^2$ = 0.27 | | N = 127 Adj. $R^2$ = 0.12 | |

Notes: Standard Error Difference in parenthesis, Variance inflation factor values are between 1.0 and 2.0.

**Table 5.** Direct and indirect gender relationship to attitudes towards statistics.

| | Direct (Step 3 $\beta$-Value, Table 4a,b) | Indirect (Difference between Step 1 and 3) | Total (Step 1, $\beta$-Value, Table 4a,b) | Significance Level, Change from Direct to Total |
|---|---|---|---|---|
| Affect | 0.41 | 0.354 | 0.056 | 6.6% to 8.2% |
| Value | 0.49 | 0.15 | 0.34 | 0.5% to 8.9% |
| Difficulty | 0.43 | 0.11 | 0.32 | 0.1% to 3.5% |
| Interest | 0.70 | 0.09 | 0.61 | 0.2% to 1.5% |
| Cognitive Competence | 0.34 | 0.338 | 0.002 | 7.4% to 99.0% |
| Effort | −0.59 | 0.38 | −0.21 | 0.2% to 29.0% |

## 5. Discussion

The results in this study regarding gender differences in attitudes towards statistics are in line and almost identical with previous investigations [27,28]. The men undergraduate respondents had more positive attitudes towards statistics. They found statistics more useful, interesting, and easier to learn than did their women peers. The women struggled more to acquire statistics and had to put significantly more effort into learning statistics. This affected the choice of pathway for further studies, according to previous research.

There was also a substantial gender difference in personal characteristics among business students in Norway. This confirmed findings from other countries [39]. Women had higher scores in the areas of Agreeableness, Conscientiousness, and Neuroticism. The gender difference remains in Norway even with greater gender equality.

The most important contribution of this article is that it simultaneously combines gender, mathematical abilities, and personal characteristics in the analysis of students' attitudes towards statistics. The investigations showed interesting results. Adjusting for personal traits and mathematical skills, the gender gap in attitudes towards statistics appeared to be substantially lower. It appears women still found statistics more difficult than did the men ($\beta = 0.32$, $p < 0.05$), while the men seemed to consider statistics to be more interesting ($\beta = 0.56$, $p < 0.05$) than did their women peers. The gender relationship of Value was significantly weak ($\beta = 0.34$, $p < 0.10$), and for the three other dimensions (Affect, Cognitive Competence, and Effort), there was no significant gender difference.

Mathematical aptitudes are a predictor of success in business statistics [52] and will have a positive influence on attitudes towards statistics [36]. Our results seem to confirm this connection. There were significant positive correlations between students with backgrounds in mathematics for natural science (N-maths), and the level of scores in Affect, Value, and Difficulty ("it is easy to learn statistics") towards statistics. Mathematics for business and social science (S-maths) includes some theoretical statistics. Since students with backgrounds in S- mathematics from upper secondary school already are familiar with business statistics, it makes sense that S-mathematics and Effort towards statistics would be significantly negatively related.

Personal characteristics are related to students' attitudes towards statistics. This study suggests that this applies especially for Conscientiousness. Hard-working and achievement-oriented students use a lot of energy in learning statistics. Therefore, there is a strong and positive link between Conscientiousness and Effort towards statistics ($\beta = 0.56$, $p < 0.01$). In fact, there is a significant positive connection between Conscientiousness and all factors of attitudes towards statistics, except for Difficulty. Conscientious students probably see the advantages of being competent in statistics to ensure good performance and to have success in their careers since Conscientiousness is a reliable predictor of academic performance [53]. This can explain the close relationship between Conscientiousness and positive attitudes towards statistics [54]. Our data suggests that students who are open to new experiences find statistics interesting ($\beta = 0.41$, $p < 0.05$). Neurotic students seemed to have bad feelings about statistics since there was a significant negative correlation between Neuroticism and Affect. There was also a substantial negative link between Neuroticism and Cognitive Competence in statistics. One might expect that such students would lose interest in statistics and reduce their effort; however, our data did not confirm this. There was no statistical link between Neuroticism and

Interest in statistics, and note also, there was a significant positive link between Neuroticism and Effort. Even those students who disliked statistics seemed to realize the subject was important. Therefore, they would spend more time learning statistics.

## 6. Limitations

The present dataset is from only one business school in Norway. It is difficult to say how valid the results are in an international context. There are probably other important factors to explain the attitudes towards statistics since the R square values are rather low.

This study applied the original version of attitudes towards statistics (SATS-36). To ensure a higher score of goodness-of-fit, some researchers conducted an explanatory factor analysis and therefore used adjusted versions of SATS-36. In line with many other papers [28,55–57], this procedure was not considered here. The original version is widely used internationally.

## 7. Conclusions

Similar to previous research, this study reported a substantial gender gap among business students in Norway in attitudes towards statistics. However, after taking into account the gender differences in personal traits and the level of mathematical skills, the gender differences in attitudes towards statistics were significantly reduced. Some personality characteristics have obviously different effects on men and women and there is also a link between mathematical skills and attitudes towards statistics.

Since business statistics is an important tool for success in business studies, students need to know more about which factors influence their attitudes towards statistics and to further explore the gender difference. Hopefully, this article has contributed to achieve more insight into this topic.

The choice of subjects and especially the mathematical pathway at upper secondary school matters. One policy implication is to ensure the students are well informed about the consequences of the various choices made at upper secondary school.

**Funding:** This research received no external funding.

**Conflicts of Interest:** The author declares no conflict of interest.

## Appendix A

Table A1 gives an overview of the sample used in this analysis. The values for Skewness and Kurtosis are well within acceptable values, except for the factor Effort. The Scala Reliabilities scores are also appropriate. If we exclude Agreeableness, all Cronbach's Alfa values are around 0.6 or higher.

Table A2 shows the correlation among the variables.

**Table A1.** Descriptive statistics, skewness, kurtosis, and scale reliability (Cronbach's Alfa).

|  | N | Min | Max | Mean | Std. Deviation | Skewness | Std. Error | Kurtosis | Std. Error | Scale Reliability Cronbach's Alfa |
|---|---|---|---|---|---|---|---|---|---|---|
| Affect | 131 | 1.00 | 7.00 | 4.5941 | 1.24911 | −0.460 | 0.212 | 0.151 | 0.420 | 0.837 |
| Cog. Competence | 131 | 1.83 | 7.00 | 5.1883 | 1.07934 | −0.516 | 0.212 | −0.053 | 0.420 | 0.840 |
| Value | 131 | 1.33 | 6.67 | 4.4448 | 0.99616 | −0.220 | 0.212 | −0.015 | 0.420 | 0.851 |
| Difficulty | 131 | 1.14 | 5.29 | 3.5217 | 0.75128 | 0.065 | 0.212 | 0.189 | 0.420 | 0.649 |
| Interest | 131 | 1.00 | 7.00 | 4.4994 | 1.29114 | −0.222 | 0.212 | −0.192 | 0.420 | 0.854 |
| Effort | 131 | 1.50 | 7.00 | 5.6698 | 1.08577 | −1.250 | 0.212 | 1.847 | 0.420 | 0.700 |
| Extraversion | 130 | 1.75 | 5.00 | 3.6404 | 0.78305 | −0.211 | 0.212 | −0.483 | 0.422 | 0.835 |
| Agreeableness | 130 | 2.25 | 5.00 | 3.9103 | 0.56933 | −0.574 | 0.212 | 0.110 | 0.422 | 0.486 |
| Conscientiousness | 130 | 1.50 | 5.00 | 3.6679 | 0.70245 | −0.569 | 0.212 | 0.223 | 0.422 | 0.711 |
| Neuroticism | 130 | 1.00 | 4.50 | 2.7013 | 0.79110 | 0.041 | 0.212 | −0.471 | 0.422 | 0.737 |
| Openness | 130 | 1.50 | 5.00 | 3.3295 | 0.72655 | −0.120 | 0.212 | −0.482 | 0.422 | 0.589 |
| Valid N (listwise) | 130 |  |  |  |  |  |  |  |  |  |

**Table A2.** Coefficient correlations.

| | 1 | 2 | 3 | 4 | 5 | 6 | 7 | 8 | 9 | 10 | 11 | 12 | 13 |
|---|---|---|---|---|---|---|---|---|---|---|---|---|---|
| 1 | - | 0.035 | −0.002 | −0.108 | 0.069 | 0.160 | −0.037 | −0.072 | −0.040 | 0.064 | −0.040 | 0.645 | −0.026 |
| 2 | 0.035 | - | 0.170 | 0.063 | 0.225 | 0.037 | −0.343 | 0.241 | 0.052 | 0.229 | −0.289 | −0.041 | −0.013 |
| 3 | −0.002 | 0.170 | - | 0.048 | 0.222 | 0.099 | 0.054 | 0.123 | −0.215 | 0.425 | −0.187 | −0.070 | −0.103 |
| 4 | −0.108 | 0.063 | 0.048 | - | −0.075 | −0.159 | −0.158 | −0.212 | 0.099 | 0.030 | 0.005 | −0.091 | 0.118 |
| 5 | 0.069 | 0.225 | 0.222 | −0.075 | - | −0.021 | −0.091 | −0.040 | −0.439 | 0.218 | −0.327 | −0.079 | −0.336 |
| 6 | 0.160 | 0.037 | 0.099 | −0.159 | −0.021 | − | −0.107 | −0.318 | 0.167 | −0.177 | −0.001 | 0.156 | −0.173 |
| 7 | −0.037 | −0.343 | 0.054 | −0.158 | −0.091 | −0.107 | - | 0.041 | −0.093 | 0.227 | 0.128 | 0.092 | 0.022 |
| 8 | −0.072 | 0.241 | 0.123 | −0.212 | −0.040 | −0.318 | 0.041 | − | −0.067 | 0.092 | −0.090 | −0.062 | 0.014 |
| 9 | −0.040 | 0.052 | −0.215 | 0.099 | −0.439 | 0.167 | −0.093 | −0.067 | − | −0.074 | 0.094 | −0.046 | 0.037 |
| 10 | 0.064 | 0.229 | 0.425 | 0.030 | 0.218 | −0.177 | 0.227 | 0.092 | −0.074 | - | −0.055 | 0.043 | −0.002 |
| 11 | −0.040 | −0.289 | −0.187 | 0.005 | −0.327 | −0.001 | 0.128 | −0.090 | 0.094 | −0.055 | - | 0.057 | −0.454 |
| 12 | 0.645 | −0.041 | −0.070 | −0.091 | −0.079 | 0.156 | 0.092 | −0.062 | −0.046 | 0.043 | 0.057 | - | −0.131 |
| 13 | −0.026 | −0.013 | −0.103 | 0.118 | −0.336 | −0.173 | 0.022 | 0.014 | 0.037 | −0.002 | −0.454 | −0.131 | - |

1. S-math. 2. Openness. 3. Gender. 4. Agreeableness. 5. Affect. 6. Effort. 7. Extraversion. 8. Conscientiousness. 9. Difficulty. 10. Neuroticism. 11. Interest. 12. N-math. 13. Value.

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
