# Peer review of "Attitudes towards Statistics among Business Students: Do Gender, Mathematical Skills and Personal Traits Matter?"

_sustainability, doi:10.3390/su12156104_

Round 1

Reviewer 1 Report

This is an interesting paper, well written, on a classical topic, but still actual.

Here are few suggestions for the improvement of the paper:

  • in the introductory section when naming other studies on the topic of gender gap among students in performance, attitudes and choice, please do provide minimum details on these papers findings.
  • also in the introductory section shortly explain The Big Five Model, the first time it is mentioned in the paper.
  • in the section on Attitudes towards Statistics you talk about the SATS-36. Clearly state that you use it in your paper. 
  • at the regression model better explain variables N-maths, S- maths. Why P-maths was not included? Some short description of the main differences between N, P and S maths in Norway. How does a pupil end up in one or another of these classes? Elective system, by specialization choice?
  • better explain what is included in Table 1c: do you talk about preferences, about attendeces ? What combination (if any) at individual level?
  • page 7, raws 212-214: move paragraph about H2 under the table that discuss it.
  • in Table 3a, some figures miss the commas
  • page 8, raw 227 editing mistake (take out TO)
  • I suggest to include the national context of Norway under both Contributions of a paper (= a new national context) an under limitations (= only one national context)
  • conclusions to be developed
  • include comments about the practical/managerial/institutional implications of the study.

Author Response

See enclosed file

Reviewer 2 Report

The research model in Figure 1 does not correspond to the content of the paper. The paper describes how gender relates all of personality, skills and attitudes. Therefore, the model corresponding to the text is a model where gender is on the left, with arrows to skills, personality as well as attitudes, and next arrows from skills and personality to attitudes.

The appropriate method to analyse such model is that of mediation, not the regression approach as in the current article. The mediation analysis should focus on distinguishing direct effects from indirect effects.

I do not understand why for some partial analyses, 3 years of data are included, in some other 2 years of data. The crucial dataset is about attitudes. Since you have no more than 1 year of attitudes data, please restrict all of your analyses to this one year.

The symbols *, **, *** stand for .05, .01 and .001 significance levels. Not for .1, .05, .01, as in this study. Please change to these standards. Using the .1 level is highly inappropriate, given the many tests that are executed.

I am surprised by the frequent use of ‘causal language’, especially since this article is about statistics education, implying the authors are very familiar with the notion that ‘correlation is no causation’. The data at hand is of correlational nature. So please avoid phrases as ‘had no significant effect’, ‘had an influence’, ‘have a positive influence’ or ‘have an impact’.

The paper is very sloppy in its references. On the first 4 pages, these are the errors I found:

  • Johnson et al., 2012 => different year in reference list
  • Chiesi & Primi, 2016 => different year in reference list
  • Nilsson & Hauff, 2018 => missing in reference list
  • Scau et al. (1995) => author name is Schau
  • Tempelar & Nijhuid, 2007 => missing in reference list, and should read: Tempelaar et al., 2007
  • Wisenbacker and Scott, (1997) => author name is Wisenbaker
  • Feingold 1994) => missing bracket

Author Response

See enclose file

Reviewer 3 Report

A very interesting paper and well written.  There are a few areas that need improvement in presenting the data and the survey forms used.  I have attached a copy of the paper with my notes.  

Author Response

See enclosed file

Round 2

Reviewer 2 Report

This is a major improvement: the mediation aspect is properly dealt with.

Mind that there are problems with your manuscript in Table 2c and the surrounding text. Part of the text got mixed up with the table; that needs repair. 

I was searching for the supplementary materials, but the link leads to the non-track changes file, where the main manuscript is the trach-changes file. The manuscript needs a supplement: descriptive statistics of the survey data that are not included in the paper itself (scale reliabilities, kurtosis and skewness) and the full correlation matrix. Those are standard components of a scholarly paper.

Author Response

See enlosed file
